# Outcomes of Nonagenarians with Acute Myocardial Infarction with or without Coronary Intervention

**DOI:** 10.3390/jcm11061593

**Published:** 2022-03-14

**Authors:** Seok Oh, Myung Ho Jeong, Kyung Hoon Cho, Min Chul Kim, Doo Sun Sim, Young Joon Hong, Ju Han Kim, Youngkeun Ahn

**Affiliations:** 1Department of Cardiology, Chonnam National University Hospital, Gwangju 61469, Korea; seokohmd87@gmail.com (S.O.); zarazoro@hanmail.net (K.H.C.); kmc3242@hanmail.net (M.C.K.); true1021@naver.com (D.S.S.); hyj200@hanmail.net (Y.J.H.); kim@zuhan.com (J.H.K.); cecilyk@hanmail.net (Y.A.); 2Department of Cardiology, Chonnam National University Medical School, Hwasun 58128, Korea

**Keywords:** myocardial infarction, nonagenarians, percutaneous coronary intervention

## Abstract

Percutaneous coronary intervention (PCI) is the mainstay treatment of acute myocardial infarction (AMI); however, many clinicians are reluctant to perform PCI in the elderly population. This study aimed to compare the clinical outcomes of PCI versus medical therapy in nonagenarian Korean patients with AMI. We compared the clinical outcomes of nonagenarian patients with AMI with or without PCI. From the pooled data, based on a series of Korean AMI registries during 2005–2020, 467 consecutive patients were selected and categorized into two groups: the PCI and no-PCI groups. The primary endpoint was 1-year major adverse cardiac event (MACE), a composite of all-cause death, non-fatal myocardial infarction, and any revascularization. Among the 467 participants, 68.5% received PCI. The PCI group had lower proportions of Killip classes III-IV, previous heart failure, and left ventricular ejection fraction <40%, but had higher proportions of all prescribed medications and STEMI diagnosis. The 1-year MACE and all-cause death were higher in the no-PCI group, although partially attenuated post-IPTW. Our study showed that nonagenarian patients with AMI undergoing PCI had better clinical outcomes than those without PCI. Nonetheless, further investigation is needed in the future to elucidate whether PCI is beneficial for this population.

## 1. Introduction

Acute myocardial infarction (AMI), an urgent or emergent medical condition, is a leading cause of morbidity and mortality worldwide [1]. The multi-faceted revolutionary innovations in pharmacological and interventional strategies have contributed to the treatment and improvement of the prognosis of AMI. However, its prevalence has gradually increased along with the trend of prolonged life expectancy. Moreover, since South Korea has become one of the world’s fastest aging nations [2] and the mean age of Korean patients with AMI has gradually increased [3], percutaneous coronary intervention (PCI) is performed more often in elderly patients. In particular, the number of nonagenarian patients with AMI will increase exponentially.

The characteristics and outcomes in this population remain poorly understood as they represent a very small portion of the overall AMI population [4,5] and are excluded from many cardiovascular clinical trials, due to various reasons such as comorbidities, impaired functional status and cognition, and limited life expectancy [6,7,8]. Generally, increased age is an independent predictor of adverse events in AMI [9,10,11]. Moreover, elderly patients with AMI have a higher burden of comorbidities. In addition, PCI is associated with increased incidences of vascular complications, bleeding complications, and cardiac death in the elderly population [12,13]. PCI is a well-established treatment strategy for AMI [14]; however, in the elderly population, such as those aged ≥90 years, many clinicians and caregivers tend to reject such invasive treatment, due to the reasons mentioned above. Moreover, it is not easy for clinicians to assess the risk–benefit balance between PCI and medical therapy in this population [15]. Although some clinical studies on PCI data on nonagenarians are available in the literature [16,17,18], there has been a paucity of domestic information on the characteristics and clinical outcomes of nonagenarian patients with AMI in the Republic of Korea.

Here, we aimed to compare the clinical outcomes of PCI versus medical therapy in nonagenarian Korean patients with AMI.

## 2. Materials and Methods

### 2.1. Study Design and Participants

The clinical information of the study participants was extracted from the Korea Acute Myocardial Infarction Registry (KAMIR)-I, KAMIR-II, Korea Working Group on Myocardial Infarction (KorMI), KAMIR-National Institute of Health (KAMIR-NIH), and KAMIR-V registries, to collect the nationwide data and standardize all clinical practices with respect to AMI in South Korea. These registries collected the data of patients with AMI from November 2005 to June 2020 and are nationwide, multicentered, and web-based prospective observational cohorts supported by the Korean Working Group of Acute Myocardial Infarction [19,20]. These prospective cohorts have their own protocols, which have been published previously [20,21]. We purposely merged all data from these registries to increase the statistical power in our study. The participating centers included 53 centers in the KAMIR-I, KAMIR-II, and KorMI registries, 20 centers in the KAMIR-NIH registry, and 43 centers in the KAMIR-V that harbor high volumes of PCI-eligible patients with facilities for PCI and on-site cardiothoracic surgery. The study protocol was approved by the institutional review board of the participating institutions. All clinical data from nonagenarian patients diagnosed with AMI were selected from the database in the present study. 

As illustrated in Figure 1, we selected 467 nonagenarian patients with AMI after excluding patients aged <90 years, those with other diagnoses than AMI, and those with invalid data. The final analysis excluded patients who died during the index hospitalization, leaving 388 nonagenarian AMI survivors. Patients were assigned to two groups depending on whether PCI was performed as follows: the PCI (*n* = 270) and no-PCI (*n* = 118) groups. 

### 2.2. Definition and Study Endpoints

The patients’ demographic and clinical characteristics were recorded. AMI was defined according to contemporary guidelines [22,23,24,25,26]. ST-segment elevation myocardial infarction (STEMI) was defined as AMI with a newly detected ST-segment elevation of ≥1 mm (0.1 mV) in ≥2 contiguous leads or newly found left bundle branch block on the 12-lead ECG. Emergency medical service (EMS) utilization refers to direct or indirect transport to a PCI-capable center through an ambulance. Off-hour visits were defined as hospital presentations during night shifts of weekdays (>6 p.m. to <8 a.m.) or weekends. The weekend refers to the period including Saturdays, Sundays, and all national public holidays in the Republic of Korea. Atypical angina refers to the inappropriate chest manifestations of typical angina. Two-dimensional echocardiography was used to evaluate the left ventricular ejection fraction (LVEF). An infarct-related artery refers to an epicardial coronary artery that was totally or partially occluded by an atheromatous or thrombotic pathologic process, which are directly responsible for acute coronary syndrome. The degree of coronary flow was quantitatively classified according to the thrombolysis in myocardial infarction (TIMI) flow grade. 

In the present study, clinical follow-up was conducted for 12 months. The primary endpoint was the occurrence of 1-year major adverse cardiac events (MACEs), which is a composite of all-cause death, non-fatal myocardial infarction (NFMI), and any revascularization. All-cause death includes both cardiac and non-cardiac death. Any revascularization is a composite of any repeat PCI and coronary artery bypass graft. The secondary endpoints included all-cause death, NFMI, and any revascularization.

### 2.3. Statistical Analysis

To explore the differences in clinical outcomes between the two groups with or without PCI, we performed statistical analysis. Student’s *t*-test was used to analyze normally distributed continuous variables. Pearson’s chi-squared test or Fisher’s two-by-two exact test were used to analyze discrete (categorical) variables. Continuous variables were described as mean ± standard deviation and discrete (categorical) variables were described as percentages with numbers. Statistical significance was set at a two-sided *p* < 0.05. 

Inverse probability of treatment weighting (IPTW) was utilized to minimize the selection bias. We constructed the propensity score using multiple logistic regression with 19 covariates, which included sex (male or female sex), Killip classification (I-II versus III-IV), body mass index, previous medical history (six items), smoking history, family history of coronary artery disease, prescribed medications (five items), thrombolysis, LVEF <40%, and STEMI diagnosis. All patients with missing covariate data were excluded from the IPTW-adjusted statistical analysis.

## 3. Results

The trends in patient volume and PCI rates in Korean nonagenarian patients with AMI are illustrated in Figure 2. The overall number of nonagenarian patients with AMI gradually increased from 76 in the KAMIR-I and KAMIR-II registries to 153 in the KAMIR-V registry. The proportion of PCI gradually increased from 46.1% in the KAMIR-I and KAMIR-II registries to 80.4 % in the KAMIR-V registry.

As explained in the study scheme in Figure 1, 467 consecutive nonagenarian patients with AMI were included in the overall analysis. The baseline characteristics of the study population are presented in Table 1. Among the overall number of participants, 68.5% underwent PCI. The PCI group had a lower proportion of Killip classes III-IV. As for previous medical history, previous heart failure was less prevalent in the PCI group than that in the no-PCI group. All medications were prescribed at a higher frequency in the PCI group. The proportion of patients with LVEF < 40% was lower in the PCI group than that in the no-PCI group. However, the PCI group had a higher proportion of patients diagnosed with STEMI. These between-group differences in the baseline characteristics were statistically balanced after IPTW adjustment (Table 1). We also investigated the clinical characteristics at the time of hospital visit of participants (Appendix A), which demonstrated similarity between the two groups in terms of EMS utilization, onset-to-door time, and off-hour presentation but differences in the prevalence of atypical anginal pain. In PCI-treated participants, coronary angiography and procedural characteristics were further investigated (Appendix A).

As for in-hospital complications (Table 2), the PCI group had a higher incidence of temporary pacemaker and intra-aortic balloon pump than that in the no-PCI group. However, CVA was more prevalent in the no-PCI group than in the PCI group. However, after IPTW adjustment, the PCI group had higher incidences of cardiogenic shock or cardiac arrest, atrioventricular block, ventricular tachycardia or fibrillation, atrial fibrillation, cardiopulmonary resuscitation, and intra-aortic balloon pump, compared to those of the no-PCI group. Among the survivors successfully discharged from the hospital, the 1-year clinical outcomes of MACE, all-cause death (cardiac death and non-cardiac death), NFMI, and any revascularization were determined, as shown in Table 3. Before IPTW adjustment, the incidence of MACE and all-cause death was higher in the no-PCI group than that in the PCI group. As these differences in MACE and all-cause death were statistically attenuated after IPTW adjustment, there were no significant differences between the two groups with the exception of any revascularization.

## 4. Discussion

Since elderly individuals constitute an increasing proportion of the overall population [27], patients over 80 years of age with AMI are expected to increase the overall AMI population. Since South Korea is an aging society, it is expected to progress toward a super-aged society in the future and can be considered one of the fastest-aging countries worldwide [2,28]. With this aging trend, the number of nonagenarian patients with AMI and their PCI rates have increased gradually in South Korea, as described in Figure 2. Nonetheless, although it is expected that the number of nonagenarian AMI patients will also gradually increase, they are under-represented in the literature concerning PCI, and many randomized controlled trials on PCI have included insufficient or negligible numbers of nonagenarian participants [29]. 

In the present study, we performed a comparative analysis of the clinical outcomes among nonagenarian patients with AMI, depending on the treatment strategy. We analyzed the clinical data of 467 consecutive patients derived from the database of the KAMIR-I, KAMIR-II, KorMI, KAMIR-NIH, and KAMIR-V registries. Our results demonstrated that the PCI group showed better 1-year clinical outcomes than the no-PCI group, with lower incidences of MACE and all-cause death. However, after IPTW adjustment, these findings were statistically attenuated, showing relatively similar outcomes in both groups.

According to the baseline characteristics described in Table 1, nonagenarian patients with AMI showed some notable features that differed from those in the general AMI population. The proportion of male patients in the two groups was 40.6% and 35.0%, respectively. According to a review article on the temporal trend of Korean patients with AMI, male patients accounted for 66.9% in 2005 and 78.0% in 2018 [3], suggesting that the proportion of female patients was relatively higher in the nonagenarian AMI population than in the general AMI population. According to a further investigation of clinical characteristics at the time of hospital visit (Appendix A), the proportion of patients with atypical chest pain was relatively high compared to the general AMI population. Patients with atypical chest pain tend to be older and females [30]. Since the mean age of the patients in the present study was over 90 years and the proportion of female patients was high, as mentioned above, this is a sufficiently predictable result. In contrast, considering that PCI was performed in 96.7% of STEMI and 82.7% of NSTEMI in the general AMI population [20], nonagenarian patients with AMI seemed to have relatively low PCI rates (68.5%), despite the temporal rise in their PCI rates, which was mentioned earlier.

In the study population, the PCI group had a higher frequency of STEMI diagnosis. Since primary PCI should be performed for timely revascularization in the case of STEMI, these findings are sufficiently predictable. Despite this finding, the PCI group had a relatively lower disease severity with respect to several clinical variables. The no-PCI group had a greater Killip functional class and lower LVEF than the PCI group. Moreover, in terms of comorbidities, the no-PCI group had a higher proportion of previous heart failure. These characteristics make interventional cardiologists reluctant to perform PCI. 

In terms of in-hospital outcomes and complications, the PCI group experienced cardiogenic shock or cardiac arrest and arrhythmic events, including atrioventricular block, ventricular tachycardia or fibrillation, and atrial fibrillation, at a higher rate. They also received cardiopulmonary resuscitation and intra-aortic balloon pumps at a higher rate. However, although the PCI group was generally more exposed to dangerous clinical situations, these findings did not contribute to a significant difference in in-hospital death in either group.

In the 1-year clinical outcomes, MACE was higher in the no-PCI group than in the PCI group. This between-group difference is driven mainly by all-cause mortality. These findings support the argument that PCI should be performed in elderly patients with AMI. Notably, the PCI group tended to receive more stringent medical treatments. According to the information on prescribed medications in Table 1, all medications including dual antiplatelet therapy, beta-blockers, angiotensin-converting enzyme inhibitors/angiotensin receptor blockers, and statins were more frequently prescribed in the PCI group than in the medical group. Thus, the PCI group received a more optimal medical treatment and appropriate reperfusion treatment. Therefore, it remains unclear whether this group benefited from an appropriate reperfusion strategy or a more optimal medical therapy. Moreover, after adjusting for many covariates, including prescribed medications, the differences in the clinical outcomes between the two groups were statistically significant. Thus, whether PCI improved MACE in the nonagenarian AMI group is controversial. As mentioned earlier, nonagenarians undergoing PCI tend to receive more optimal medical therapy, and it is suggested that when PCI was not considered for various reasons, we also tend to pay lesser attention to medical treatment in this population. 

In the multivariable logistic regression analysis, both Killip class III-IV and diabetes mellitus were found as predictors for MACE. High Killip class is an independent predictor of mortality in acute coronary syndrome [31,32]. Type 2 diabetes mellitus is also well-known as a risk factor related to the development of cardiovascular disorders [33,34], which increases inflammatory process and promotes or deteriorate vascular remodeling [35].

We further investigated the angiographic and procedural characteristics in the PCI group to evaluate whether they received high-quality PCI (Appendix A). Approximately two-thirds of the patients with AMI received a transfemoral approach, although current guidelines recommend the radial approach over the femoral approach in patients with AMI [36]. Since interventional cardiologists with high proficiency with radial route tend to be associated with worse outcomes of PCI via femoral artery [37], they should be also encouraged to increase their proficiency in PCI via femoral artery, given the high application of femoral artery in the elderly population. During PCI, the stent implantation rate was 85.1%, and most patients received drug-eluting stent implantation (71.3%). Since the current guidelines recommend drug-eluting stents over bare-metal stents in any PCI [22], and drug-eluting stents seems to be associated with lower long-term mortality in comparison to bare-metal stents [38], it can be considered an appropriate finding. Although 62.1% had pre-PCI TIMI 0-I, the proportion of post-PCI TIMI II-III was 97.1%, with a success rate of 95.9%. Therefore, it can be seen that nonagenarian patients with AMI receive high-quality PCI. In addition, their mortality rate seems to be higher than that of the general AMI population [3]; however, it is comparable to that of the no-PCI group. 

Through literature review, several foreign studies on clinical outcomes of nonagenarian patients with AMI undergoing PCI were found [18,39]. Most nonagenarians undergoing PCI have a high-risk profile with a greater burden of comorbidities [18,29]; however, PCI is a feasible and safe procedure in nonagenarians with accepTable 3-year survival rates [17,18]. In special situations, such as STEMI, primary PCI is emphasized as a mainstay treatment and should be performed in a timely and routine manner [40]. Advanced age is not an absolute contraindication to PCI; however, many clinicians tend to be reluctant to perform PCI in this population. Since there are insufficient data with limited quality on the risk–benefit and cost–benefit profiles of PCI procedures in the very elderly population, there have been arguments in favor of and against the implementation of PCI for them [15]. Moreover, several foreign studies emphasize that nonagenarians have higher rates of in-hospital complications and worse outcomes than younger counterparts [16,41]. Domestic data on PCI in nonagenarian patients with AMI are still very scarce. According to an article by Kim et al., the number of nonagenarian patients with STEMI undergoing primary PCI tends to increase in the Republic of Korea, with a high success rate and an acceptable in-hospital mortality rate [42]. In a comparative analysis of octogenarian versus nonagenarian patients with AMI, nonagenarian patients had similar incidences of 1-year MACE compared to octogenarian patients, and PCI was associated with better 1-year clinical outcomes [4]. In a cohort study, mortality after AMI was reduced in correlation with the PCI procedure in the nonagenarian AMI population [43]. However, since this result was based on a single-centered experience with a small study cohort, it is difficult to generalize with low statistical power. In contrast, the present study was conducted by the database from nationwide Korean multicenter observational cohorts, the KAMIR-I, KAMIR-II, KorMI, KAMIR-NIH, and KAMIR-V registries. To the best of our knowledge, our study is the first multicenter comparative study of PCI versus no-PCI in Korean nonagenarian patients with AMI.

This study has some limitations when interpreting the results of this study. The participating institutions in the KAMIR, KorMI, and KAMIR-NIH registries were tertiary cardiovascular institutions with higher volumes of patients with AMI and higher annual PCI rates than average medical centers. Hence, it is difficult to generalize the clinical patterns and outcomes of our study. In addition, as this was not a randomized study, there were some problems pertaining to selection bias. Statistical adjustment using IPTW was performed to overcome this limitation; however, a large-scale multicentered randomized controlled trial must be conducted in the future to draw in-depth conclusions on the clinical outcomes of PCI in the nonagenarian AMI population. Moreover, this may be considered a notable analysis of nonagenarian patients with AMI; however, most participants were enrolled and analyzed before the COVID-19 pandemic. Since there are expected characteristic differences in the clinical features, treatment quality, and outcomes of the AMI population during the COVID-19 pandemic, further studies are needed. In addition, since Korea is one of the fastest-aging countries, further prospective and systematic research on nonagenarian AMI populations is needed in the future.

## 5. Conclusions

Nonagenarian patients with AMI who underwent PCI may have better clinical outcomes than those treated with medical therapy only, although it is relatively more dangerous during the initial hospitalization. Nevertheless, whether PCI is beneficial in this age group remains inconclusive. Therefore, further investigation is needed in the future.

Key message

Nonagenarian patients with AMI who underwent PCI appeared to have better clinical outcomes than those who were conservatively treated with no-PCI.Nonagenarian patients with AMI who underwent PCI also tended to receive more optimal medical therapy than those who were conservatively treated with no-PCI.Nonagenarian patients with AMI received high-quality PCI. Despite dangerous post-PCI clinical situations, in-hospital death was comparable in both groups.In routine practice, many clinicians are still reluctant to implement PCI in nonagenarian patients with AMI.Whether PCI is truly beneficial in nonagenarian AMI patients remains controversial, and further investigation is needed in the future.

## Figures and Tables

**Figure 1 jcm-11-01593-f001:**
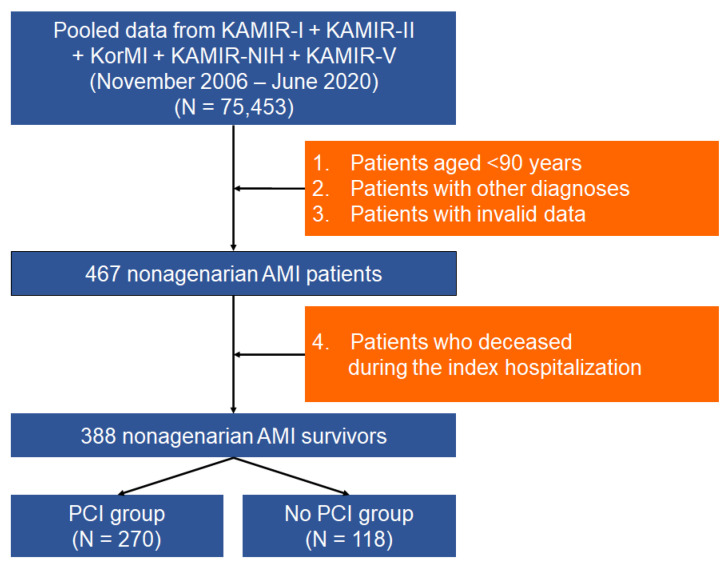
Diagram of the study design. AMI, acute myocardial infarction; KAMIR, Korean Acute Myocardial Infarction Registry; KAMIR-NIH, Korean Acute Myocardial Infarction Registry-National Institute of Health; KorMI, Korea Working Group on Myocardial Infarction; PCI, percutaneous coronary intervention.

**Figure 2 jcm-11-01593-f002:**
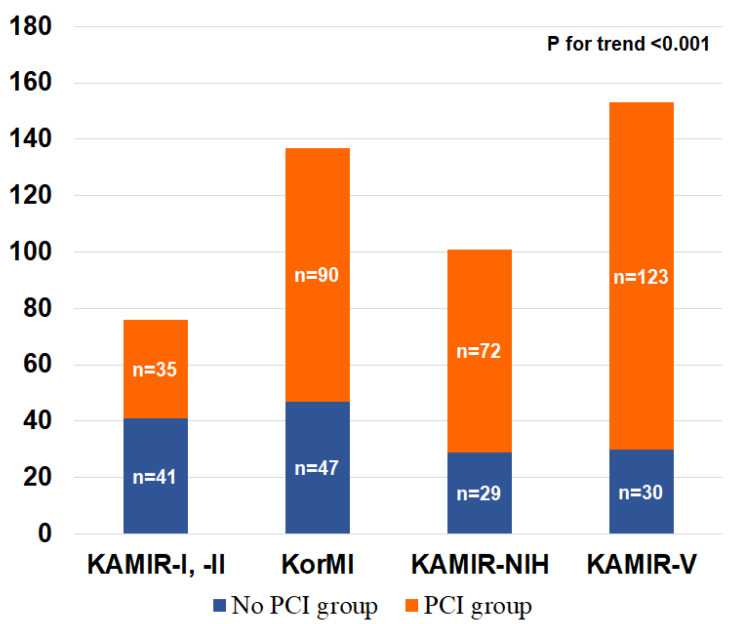
Temporal trends in the PCI/no-PCI ratio among Korean nonagenarian patients with AMI. AMI, acute myocardial infarction; KAMIR, Korean Acute Myocardial Infarction Registry; KAMIR-NIH, Korean Acute Myocardial Infarction Registry-National Institute of Health; KorMI, Korea Working Group on Myocardial Infarction; PCI, percutaneous coronary intervention.

**Table 1 jcm-11-01593-t001:** Baseline characteristics of the patients.

	Before IPTW Adjustment			After IPTW Adjustment		
Characteristics	PCI Group	No-PCI Group	*p*-Value	PCI Group	No-PCI Group	*p*-Value
	(*n* = 320)	(*n* = 147)		(*n* = 286)	(*n* = 265)	
Male patients	134 (41.9)	48 (32.7)	0.058	121 (42.5)	108 (40.7)	0.885
Killip class III-IV	69 (22.0)	57 (42.5)	<0.001	74 (26.0)	59 (22.2)	0.645
BMI ≥ 25 kg/m^2^	41 (15.4)	22 (21.6)	0.156	45 (15.7)	33 (12.6)	0.629
Previous medical history						
Hypertension	208 (65.2)	82 (56.6)	0.074	189 (66.1)	197 (74.3)	0.376
Diabetes mellitus	49 (15.4)	29 (19.9)	0.233	49 (17.1)	56 (21.3)	0.677
Dyslipidemia	18 (5.6)	8 (5.7)	0.989	16 (5.7)	7 (2.5)	0.236
Ischemic heart disease	50 (15.6)	22 (15.0)	0.855	47 (16.3)	54 (20.4)	0.673
Previous heart failure	14 (4.4)	14 (9.5)	0.030	12 (4.2)	12 (4.4)	0.919
Old CVA	22 (6.9)	14 (9.5)	0.319	21 (7.5)	12 (4.6)	0.436
Smoking history	87 (28.2)	40 (28.0)	0.952	94 (32.9)	122 (45.9)	0.282
Family CAD history	11 (3.7)	6 (4.9)	0.569	23 (8.1)	35 (13.2)	0.589
Prescribed medications						
Aspirin	285 (89.1)	112 (76.2)	<0.001	270 (94.5)	208 (78.6)	0.065
P2Y12 inhibitors	285 (89.1)	92 (62.6)	<0.001	271 (94.8)	242 (91.4)	0.339
Beta-blockers	191 (59.7)	59 (40.1)	<0.001	180 (63.1)	119 (44.8)	0.132
ACEIs/ARBs	201 (62.8)	65 (44.2)	<0.001	184 (64.3)	114 (42.9)	0.074
Statins	240 (75.0)	73 (49.7)	<0.001	218 (76.2)	162 (61.0)	0.214
LVEF < 40%	72 (26.3)	42 (37.8)	0.024	98 (34.3)	120 (45.2)	0.373
STEMI diagnosis	202 (63.1)	41 (27.9)	<0.001	141 (49.5)	103 (39.0)	0.413

Values are presented as number (percentage) for categorical values and means ± standard deviation for continuous variables. ACEI, angiotensin-converting enzyme inhibitor; ARB, angiotensin receptor blocker; BMI, body-mass index; CAD, coronary artery disease; CVA, cerebrovascular accident; IPTW, inverse probability of treatment weighting; LVEF, left ventricular ejection fraction; PCI, percutaneous coronary intervention; STEMI, ST-segment elevation myocardial infarction.

**Table 2 jcm-11-01593-t002:** In-hospital complications.

	Before IPTW Adjustment			After IPTW Adjustment		
Characteristics	PCI Group	No-PCI Group	*p*-Value	PCI Group	No-PCI Group	*p*-Value
	(*n* = 320)	(*n* = 147)		(*n* = 286)	(*n* = 265)	
Cardiogenic shock or cardiac arrest	48 (15.0)	18 (12.2)	0.427	32 (11.2)	10 (3.6)	0.026
New-onset heart failure	24 (7.5)	14 (9.5)	0.458	37 (13.0)	57 (21.4)	0.333
Re-occurring MI	2 (0.6)	0 (0.0)	1.000	0 (0.0)	0 (0.0)	1.000
CVA	4 (1.3)	7 (4.8)	0.042	3 (1.1)	5 (1.8)	0.539
Atrioventricular block	11 (3.4)	3 (2.0)	0.564	6 (2.1)	0 (0.0)	0.041
Ventricular tachycardia or fibrillation	12 (3.8)	2 (1.4)	0.243	9 (3.3)	0 (0.0)	0.023
Atrial fibrillation	16 (5.0)	4 (2.7)	0.330	17 (6.1)	1 (0.5)	0.003
Acute kidney injury	6 (1.9)	6 (4.1)	0.162	10 (3.6)	4 (1.7)	0.426
Sepsis	4 (1.3)	1 (0.7)	1.000	10 (3.6)	1 (0.5)	0.056
Multi-organ failure	6 (1.9)	4 (2.7)	0.515	8 (2.7)	2 (0.9)	0.254
Temporary pacemaker	34 (10.6)	3 (2.0)	0.001	22 (7.7)	5 (2.0)	0.059
Cardiopulmonary resuscitation	35 (10.9)	10 (6.8)	0.160	33 (11.5)	8 (3.1)	0.021
Intra-aortic balloon pump	15 (4.7)	1 (0.7)	0.027	11 (3.9)	0 (0.0)	0.018
Defibrillation	8 (2.5)	0 (0.0)	0.061	5 (1.8)	0 (0.0)	0.126
In-hospital death	50 (15.6)	29 (19.7)	0.272	28 (9.8)	15 (5.7)	0.308

Values are presented as number (percentage) for categorical values and means ± standard deviation for continuous variables. CVA, cerebrovascular accident; IPTW, inverse probability of treatment weighting; MI, myocardial infarction; PCI, percutaneous coronary intervention.

**Table 3 jcm-11-01593-t003:** One-year clinical outcomes.

	Before IPTW Adjustment			After IPTW Adjustment		
Characteristics	PCI Group	No-PCI Group	*p*-Value	PCI Group	No-PCI Group	*p*-Value
	(*n* = 270)	(*n* = 118)		(*n* = 258)	(*n* = 250)	
MACE	45 (16.7)	30 (25.4)	0.044	54 (21.0)	104 (41.4)	0.082
All-cause death	41 (15.2)	28 (23.7)	0.043	50 (19.6)	102 (40.8)	0.068
Cardiac death	28 (10.4)	19 (16.1)	0.111	37 (14.4)	59 (23.8)	0.363
Non-cardiac death	13 (4.8)	9 (7.6)	0.270	13 (5.2)	43 (17.0)	0.054
NFMI	6 (2.2)	1 (0.8)	0.680	9 (3.6)	2 (0.6)	0.093
Any revascularization	6 (2.2)	1 (0.8)	0.680	6 (2.3)	0 (0.0)	0.036

Values are presented as number (percentage) for categorical values and means ± standard deviation for continuous variables. IPTW, inverse probability of treatment weighting; MACE, major adverse cardiac event; NFMI, non-fatal myocardial infarction; PCI, percutaneous coronary intervention. All variables mentioned in the baseline characteristics of patients (Table 1) were included in the multivariable logistic regression analysis. Of these items, the significant predictive factors were Killip class III-IV and diabetes mellitus (Table 4).

**Table 4 jcm-11-01593-t004:** Multivariable analysis for predictors of MACE.

Univariable Logistic Analysis			Multivariable Logistic Analysis		
	Odds Ratio (95% CI)	*p*-Value		Odds Ratio (95% CI)	*p*-Value
Male patients	1.352 (0.814–2.245)	0.244	Male patients		
Killip class III-IV	1.592 (0.907–2.796)	0.105	Killip class III-IV	1.592 (0.907–2.796)	0.046
BMI ≥ 25 kg/m^2^	0.868 (0.411–1.834)	0.711	BMI ≥ 25 kg/m^2^		
Previous medical history			Previous medical history		
Hypertension	1.184 (0.699–2.007)	0.530	Hypertension		
Diabetes mellitus	2.050 (1.114–3.711)	0.021	Diabetes mellitus	2.127 (1.045–4.331)	0.037
Dyslipidemia	1.718 (0.643–4.590)	0.281	Dyslipidemia		
Ischemic heart disease	1.367 (0.706–2.648)	0.354	Ischemic heart disease		
Previous heart failure	0.872 (0.288–2.642)	0.808	Previous heart failure		
Old CVA	0.960 (0.380–2.423)	0.931	Old CVA		
Smoking history	0.881 (0.498–1.556)	0.662	Smoking history		
Family CAD history	1.213 (0.325–4.529)	0.774	Family CAD history		
Prescribed medications			Prescribed medications		
Aspirin	1.212 (0.402–3.656)	0.733	Aspirin		
P2Y12 inhibitors	1.542 (0.626–3.800)	0.347	P2Y12 inhibitors		
Beta-blockers	0.769 (0.460–1.287)	0.318	Beta-blockers		
ACEIs/ARBs	0.559 (0.333–0.939)	0.028	ACEIs/ARBs		
Statins	0.846 (0.466–1.535)	0.582	Statins		
LVEF < 40%	1.448 (0.828–2.534)	0.195	LVEF < 40%		
STEMI diagnosis	0.712 (0.428–1.183)	0.190	STEMI diagnosis		

ACEI, angiotensin-converting enzyme inhibitor; ARB, angiotensin receptor blocker; BMI, body-mass index; CAD, coronary artery disease; CI, confidence interval; CVA, cerebrovascular accident; LVEF, left ventricular ejection fraction; PCI, percutaneous coronary intervention; STEMI, ST-segment elevation myocardial infarction.

## Data Availability

Not applicable.

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
