# Peer review of "Outcomes of Nonagenarians with Acute Myocardial Infarction with or without Coronary Intervention"

_jcm, 2022, doi:10.3390/jcm11061593_

Round 1
Reviewer 1 Report
Dear Authors,
I have read with great interest your original article entitled “Outcomes of nonagenarians with acute myocardial infarction with or without coronary intervention”. I would like to congratulate you for the whole project. Briefly, data about PCI in AMI nonagenarians are poor in existing literature. The authors try to investigate whether the interventional or conservative approach will benefit such patients in one-year follow-up.
Our comments are as below:
- This retrospective study included patients from 2005 until 2020. During this period, the definitions for ACS were reconsidered; thus, a global definition cannot exist. The authors could be more comprehensive, by presenting the different definitions for each period.
- Findings from Supplementary Tables 1 and 2 should be referred firstly in “Results” section. Then, the authors could comment on them on “Discussion” section.
- In addition, the authors do not present how the patients distributed during the 15-years duration. Although they refer from which registry the data were obtained, it remains unclear the patients treated invasively/conservatively per year.
- Considering the large number of nonagenarians with AMI included in this study, the authors could perform multi-regression analysis, in order to identify which comorbidities, symptoms, etc. consist major risk factor for morbidity and mortality in such patients.
- Furthermore, authors have focused on Korean data and studies. It could be interesting if they could compare their findings with other national registries.
- Minor grammatical and syntactical errors should be corrected prior any other consideration.
Taking all the above into consideration, we believe that it is a well-written and comprehensive manuscript, depicting accurately the main findings and commenting on it adequately. Whether the topic cannot be considered as novel, data regarding this age subgroup are poor. In our opinion, the manuscript should be majorly revised prior any other consideration.
Kind regards,

Author Response
Journal Revision Letter
2 March 2022
Dear Journal of Clinical Medicine team,
Dear Editor
Thank you for your very kind and careful review of our paper, and for the comments and suggestions. It is with excitement that I re-submit to you our revised version of manuscript (manuscript ID: jcm-1617698), “Outcomes of nonagenarians with acute myocardial infarction with or without coronary intervention” for the Journal of Clinical Medicine. A revision of this paper has been carried out to take all of them into account. In this process, we believe that this paper has been significantly improved.
I have responded specifically to each suggestion below.
Dear Authors,
I have read with great interest your original article entitled “Outcomes of nonagenarians with acute myocardial infarction with or without coronary intervention”. I would like to congratulate you for the whole project. Briefly, data about PCI in AMI nonagenarians are poor in existing literature. The authors try to investigate whether the interventional or conservative approach will benefit such patients in one-year follow-up.
Our comments are as below:
- This retrospective study included patients from 2005 until 2020. During this period, the definitions for ACS were reconsidered; thus, a global definition cannot exist. The authors could be more comprehensive, by presenting the different definitions for each period.
Reply) Thank you for your good and valuable comments. I totally agree with your opinion. The definition of AMI has been constantly revised through a series of consensus document. For this reason, we would like to briefly introduce the definition of AMI as follows.
“AMI was defined according to contemporary guidelines”
- Findings from Supplementary Tables 1 and 2 should be referred firstly in “Results” section. Then, the authors could comment on them on “Discussion” section.
Reply) Thank you for your good comments. In the Result section, we mentioned as follows.
“We also investigated the clinical characteristics at the time of hospital visit of participants (Table S1), which demonstrated similarity between the two groups in terms of EMS utilization, onset-to-door time, and off-hour presentation but differences in the prevalence of atypical anginal pain. In PCI-treated participants, coronary angiography and procedural characteristics were further investigated (Table S2).”
- In addition, the authors do not present how the patients distributed during the 15-years duration. Although they refer from which registry the data were obtained, it remains unclear the patients treated invasively/conservatively per year.
Reply) Thank you for your valuable comments. Actually speaking, although we were able to demonstrate annual (year-over-year) trends in the number of nonagenarian AMI patients, we did not because the two year periods, 2005 and 2020, were different (too short) from those of the other years. In other words, the pooled analysis (KAMIR-I, II, KorMI (KAMIR-III), KAMIR-NIH, and KAMIR-V) collected patients starting in November 2005 and ending in June 2020.
- Considering the large number of nonagenarians with AMI included in this study, the authors could perform multi-regression analysis, in order to identify which comorbidities, symptoms, etc. consist major risk factor for morbidity and mortality in such patients.
Reply) Thank you for your valuable comments. In order to identify the risk factors for MACE in nonagenarian AMI patients, we commenced the multivariable analysis for predictors of MACE. After all variables mentioned in the baseline characteristics of the patients (Table 1) were included in the multivariable logistic regression analysis, both Killip class III-IV and diabetes mellitus were identified as predictors for MACE, as described in Table 4.
Therefore, we added the following paragraph to the Results and the Discussion sections, respectively.
“All variables mentioned in the baseline characteristics of patients (Table 1) were included in the multivariable logistic regression analysis. Of these items, the significant predictive factors were Killip class III-IV and diabetes mellitus (Table 4).” (in the Results section)
“In the multivariable logistic regression analysis, both Killip class III-IV and diabetes mellitus were found as predictors for MACE. High Killip class is an independent predictor of mortality in acute coronary syndrome. Type 2 diabetes mellitus is also well-known as a risk factor related to the development of cardiovascular disorders, which increases inflammatory process and promotes or deteriorate vascular remodeling.” (in the Discussion section)
- Furthermore, authors have focused on Korean data and studies. It could be interesting if they could compare their findings with other national registries.
Reply) Thank you for your valuable and insightful comments. Since the clinical outcomes of patients with nonagenarian AMI is one of the interesting topics, clinical studies have been conducted on this topic in many countries. Some studies mentioned that PCI procedure is indeed a feasible and safe procedure in nonagenarian with acceptable 3-year survival rates. However, several studies also emphasize that nonagenarians have higher rates of in-hospital complications and worse outcomes than younger counterparts. Therefore, we mentioned as follows in the Discussion section.
“Through literature review, several foreign studies on clinical outcomes of nonagenarian patients with AMI undergoing PCI were found. Most nonagenarians undergoing PCI have a high-risk profile with a greater burden of comorbidities; however, PCI is a feasible and safe procedure in nonagenarians with acceptable 3-year survival rates. In special situations, such as STEMI, primary PCI is emphasized as a mainstay treatment and should be performed in a timely and routine manner. Advanced age is not an absolute contraindication to PCI; however, many clinicians tend to be reluctant to perform PCI in this population. Since there are insufficient data with limited quality on the risk-benefit and cost-benefit profiles of PCI procedures in the very elderly population, there have been arguments in favor of and against the implementation of PCI for them. Moreover, several foreign studies emphasize that nonagenarians have higher rates of in-hospital complications and worse outcomes than younger counterparts”
- Minor grammatical and syntactical errors should be corrected prior any other consideration.
Reply) Thank you for your valuable comments. The revised manuscript went through the English editing process again (Editage).
Taking all the above into consideration, we believe that it is a well-written and comprehensive manuscript, depicting accurately the main findings and commenting on it adequately. Whether the topic cannot be considered as novel, data regarding this age subgroup are poor. In our opinion, the manuscript should be majorly revised prior any other consideration.
Kind regards,
The manuscript has certainly benefited from these insightful revision suggestions. I look forward to working with you and the reviewers to move this manuscript closer to publication in the Journal of Clinical Medicine.
Thank you for your consideration. I look forward to hearing from you.
Sincerely,
Seok Oh, MD
Department of Cardiology, Chonnam National University Hospital
42, Jebong-ro, Dong-gu, Gwangju, 61469, Korea
Tel: +82-10-6646-1690
E-mail: seokohmd87@gmail.com, seokohmd@gmail.com seokohmd@naver.com

Reviewer 2 Report
Interesting study, however some issues need some drawbacks.
Authors anlyzed relatively old data from 2005 to 2020 with trealtively low sample size. In this time there was both DES, BVS and BMS used. Any information about used stent type and possible implication? Please add in discussion this article DOI: 10.1002/ccd.25169
Echocardiographic parameters? Ejection fraction?
Aniography data? Please add inforamtion about coronary artery with culprit leasion, information about complet/incomplete revascularization
Mechancal thrombectomy utilization? GP IIb/IIIa inhibitors utilization?
How many procedures were performed with radial access? Was operator radial access evaluated? please discuss this topic in light of this publication DOI: 10.3390/jcm8091484
What was the time of procedure? Some authors suggest impact of on/off-hours on clinical outcomes, with potentially worst outcomein proceduresconducted during night shift. Please dicuss this topic DOI: 10.1016/j.hjc.2021.01.011
Finally, much larger studies presented similar results, what is new ? What this study add to current knowledge?
Author Response
Journal Revision Letter
2 March 2022
Dear Journal of Clinical Medicine team,
Dear Editor
Thank you for your very kind and careful review of our paper, and for the comments and suggestions. It is with excitement that I re-submit to you our revised version of manuscript (manuscript ID: jcm-1617698), “Outcomes of nonagenarians with acute myocardial infarction with or without coronary intervention” for the Journal of Clinical Medicine. A revision of this paper has been carried out to take all of them into account. In this process, we believe that this paper has been significantly improved.
I have responded specifically to each suggestion below.
Interesting study, however some issues need some drawbacks.
Authors anlyzed relatively old data from 2005 to 2020 with trealtively low sample size. In this time there was both DES, BVS and BMS used. Any information about used stent type and possible implication?
Reply) Thank you for your valuable and insightful comments. Actually, we also summarized the coronary angiographic and procedural characteristics in PCI-treated nonagenarian AMI patients, in Table S2 (supplementary table). According to Table S1 data, 85.1% received stent implantation with 71.3% for drug-eluting stents and 13.8% for bare-metal stents. In other words, we present information about stent types of all participants in Table S2.
Please add in discussion this article DOI: 10.1002/ccd.25169
Reply) Thank you for your valuable comment. Dr. Siudak and his colleagues emphasized that DES demonstrated lower long-term mortality in comparison to BES among patients with STEMI (DOI: 10.1002/ccd.25169). Furthermore, contemporary AMI guidelines recommend DES over BMS in any PCI procedure. For these reasons, I would like to add the following sentences in the Discussion section, citing this article.
“Since the current guidelines recommend drug-eluting stents over bare-metal stents in any PCI, and drug-eluting stents seems to be associated with lower long-term mortality in comparison to bare-metal stents, it can be considered an appropriate finding.”
Echocardiographic parameters? Ejection fraction?
Reply) Thank you for your valuable and insightful comments. Although we did not comment on the values of LVEF (i.e., mean +/- standard deviation) in the present study, a comparison of the two groups for the proportions of LVEF <40% was performed. Their results are summarized in Table 1.
Aniography data? Please add inforamtion about coronary artery with culprit leasion, information about complet/incomplete revascularization
Reply) Thank you for your valuable and insightful comments. As mentioned earlier, all angiographic profiles for the PCI-treated population are summarized in Table S2. As for information on infarct-related artery (i.e., culprit lesion), LMCA was 2.2%, LAD was 48.9%, LCX was 13.2% and RCA was 35.7%. In PCI results, 95.9% received successful PCI whereas 4.1% received suboptimal or failed PCI.
Mechancal thrombectomy utilization? GP IIb/IIIa inhibitors utilization?
Reply) Thank you for your valuable and insightful comments. As mentioned earlier, all angiographic profiles for the PCI-treated population are summarized in Table S2. The utilization rate of GPIIb/IIIa inhibitors was 8.1%.
Meanwhile, a total of 4 patients in the study population underwent thrombolysis. Of these, 2 were in the PCI group, and the other 2 were in the non-PCI group. Since the present study focused on clinical outcomes depending on whether or not PCI was performed, and the number of patients undergoing thrombolysis was extremely small (4), these information on thrombolysis were not included in the manuscript.
How many procedures were performed with radial access? Was operator radial access evaluated?
Reply) Thank you for your valuable and insightful comments. As mentioned earlier, all angiographic profiles for the PCI-treated population are summarized in Table S2. 66.8% received transfemoral approach, while others received transfemoral approach.
please discuss this topic in light of this publication DOI: 10.3390/jcm8091484
Reply) Thank you for your valuable and insightful comments. In the present study, approximately two-thirds of patients with AMI received a transfemoral approach, although current guidelines recommend a radial approach over the femoral approach in patients with AMI, as mentioned in the manuscript. Nevertheless, according to a Polish study conducted by Tokarek and his colleagues, the femoral approach seems to have several advantages compared to the radial approach with relatively low risk of periprocedural death, stroke and bleeding complications. However, there are also found studies that are more favorable to the radial approach. Li et al. demonstrated that transradial approach in STEMI patients undergoing primary PCI with DESs was associated with lower incidence of access site hematoma, 12-month repeat revascularization, and MACE compared to transfemoral approach.
For these reasons, I would like to add the following sentences in the Discussion section, citing this article.
“We further investigated the angiographic and procedural characteristics in the PCI group to evaluate whether they received high-quality PCI (Table S2). Approximately two-thirds of the patients with AMI received a transfemoral approach, although current guidelines recommend the radial approach over the femoral approach in patients with AMI. Tokarek and his colleagues showed several advantages of the femoral approach over the radial approach, with relatively low incidencese of periprocedural complications. However, there are also studies that are more favorable to the radial approach. Further research is needed on the tendency of high utilization rates of the femoral approach in this extremely elderly population.”
What was the time of procedure? Some authors suggest impact of on/off-hours on clinical outcomes, with potentially worst outcomein proceduresconducted during night shift. Please dicuss this topic DOI: 10.1016/j.hjc.2021.01.011
Reply) Thank you for your valuable and insightful comments. I totally agree with your opinion. The clinical courses or outcomes of AMI patients according to the timing of hospital visits or PCI procedures is one of the most important topics. According to a clinical study based on a Polish nationwide registry by Tokarek and his colleagues, off-hours PCI seems to be associated with higher periprocedural mortality. In South Korea (https://doi.org/10.3904/kjim.2021.204), there were found no significant differences in the long-term and short-term outcomes of STEMI undergoing PCI regardless of the time of presentation. In fact, we mentioned information on the time of hospital presentation in Table S1. (i.e., off-hour presentation) Unfortunately, information about PCI timing is not described. The reason is the imbalance in the information of the initial balloon time in the KAMIR series. For example, KAMIR-IV has a wealth of information about the initial balloon time of AMI patients who received PCI. However, there is very little information about them in KAMIR-I, II, and KorMI (i.e. KAMIR-III). In other words, we have not able to comment further on this interesting topic like off-hour PCI because of this reason. (i.e, we only mentioned the timing of hospital visit (off hour versus on-hour presentation))
Finally, much larger studies presented similar results, what is new ? What this study add to current knowledge?
Reply) Thank you for your good comments. As mentioned in the manuscript, the present study is the first multicenter comparative study of PCI versus no-PCI in Korean nonagenarian patients with AMI. Interestingly, nonagenarian patients with AMI who underwent PCI also tended to received more optimal medical therapy, than those who were conservatively treated with no-PCI. In other words, it suggests that it should not be overlooked to maintain optimal medical therapy, even for patients who have decided not to undergo PCI.
The manuscript has certainly benefited from these insightful revision suggestions. I look forward to working with you and the reviewers to move this manuscript closer to publication in the Journal of Clinical Medicine.
Thank you for your consideration. I look forward to hearing from you.
Sincerely,
Seok Oh, MD
Department of Cardiology, Chonnam National University Hospital
42, Jebong-ro, Dong-gu, Gwangju, 61469, Korea
Tel: +82-10-6646-1690
E-mail: seokohmd87@gmail.com, seokohmd@gmail.com seokohmd@naver.com

Round 2
Reviewer 1 Report
The authors have considered our proposals.
Author Response
The authors have considered our proposals.
Reply) Thank you for your kind and valuable comments.
Reviewer 2 Report
Thank you for answers. One issue need clarification. Authors relpied:
“ Tokarek and his colleagues showed several advantages of the femoral approach over the radial approach, with relatively low incidencese of periprocedural complications. However, there are also studies that are more favorable to the radial approach. Further research is needed on the tendency of high utilization rates of the femoral approach in this extremely elderly population.”
in this study RA is presented as more beneficial. However, the more procedures are performed via RA, the most risk for complication is expected when FA is used. Please, see data from summary of this study:
"Operators with the highest proficiency in RA were associated with increased risk of periprocedural death, stroke and bleeding complications at access site during angiography via FA. Similarly, higher prevalence of periprocedural mortality during PCI with FA was observed in most experienced radial operators as compared to other groups. The detrimental effect of FA utilization by the most experienced radial operators was observed in both stable angina and acute coronary syndromes. Higher experience and utilization of RA might be linked to worse outcomes of PCIs performed via femoral artery in both stable and acute settings. "
please revise yor manuscrpt accroding to real message from this study
Author Response
Thank you for answers. One issue need clarification. Authors relpied:
“ Tokarek and his colleagues showed several advantages of the femoral approach over the radial approach, with relatively low incidencese of periprocedural complications. However, there are also studies that are more favorable to the radial approach. Further research is needed on the tendency of high utilization rates of the femoral approach in this extremely elderly population.”
in this study RA is presented as more beneficial. However, the more procedures are performed via RA, the most risk for complication is expected when FA is used. Please, see data from summary of this study:
"Operators with the highest proficiency in RA were associated with increased risk of periprocedural death, stroke and bleeding complications at access site during angiography via FA. Similarly, higher prevalence of periprocedural mortality during PCI with FA was observed in most experienced radial operators as compared to other groups. The detrimental effect of FA utilization by the most experienced radial operators was observed in both stable angina and acute coronary syndromes. Higher experience and utilization of RA might be linked to worse outcomes of PCIs performed via femoral artery in both stable and acute settings. "
please revise yor manuscrpt accroding to real message from this study
Reply) Thank you for your good comments. Reflecting your opinion, I have revised the manuscript as follows.
“We further investigated the angiographic and procedural characteristics in the PCI group to evaluate whether they received high-quality PCI (Table S2). Approximately two-thirds of the patients with AMI received a transfemoral approach, although current guidelines recommend the radial approach over the femoral approach in patients with AMI. Since interventional cardiologists with high proficiency with radial route tend to be associated with worse outcomes of PCI via femoral artery, they should be also encouraged to increase their proficiency in PCI via femoral artery, given the high application of femoral artery in the elderly population.”
